# Effects of Butyrate Supplementation on Inflammation and Kidney Parameters in Type 1 Diabetes: A Randomized, Double-Blind, Placebo-Controlled Trial

**DOI:** 10.3390/jcm11133573

**Published:** 2022-06-21

**Authors:** Ninna H. Tougaard, Marie Frimodt-Møller, Hanne Salmenkari, Elisabeth B. Stougaard, Andressa D. Zawadzki, Ismo M. Mattila, Tine W. Hansen, Cristina Legido-Quigley, Sohvi Hörkkö, Carol Forsblom, Per-Henrik Groop, Markku Lehto, Peter Rossing

**Affiliations:** 1Steno Diabetes Center Copenhagen, 2730 Herlev, Denmark; marie.frimodt-moeller@regionh.dk (M.F.-M.); elisabeth.buur.stougaard@regionh.dk (E.B.S.); andressa.de.zawadzki@regionh.dk (A.D.Z.); ismo.matias.mattila@regionh.dk (I.M.M.); tine.willum.hansen@regionh.dk (T.W.H.); cristina.legido.quigley@regionh.dk (C.L.-Q.); peter.rossing@regionh.dk (P.R.); 2Department of Nephrology, Herlev University Hospital, 2730 Herlev, Denmark; 3Folkhälsan Institute of Genetics, Folkhälsan Research Center, 00290 Helsinki, Finland; hanne.salmenkari@helsinki.fi (H.S.); carol.forsblom@hus.fi (C.F.); per-henrik.groop@helsinki.fi (P.-H.G.); markku.lehto@helsinki.fi (M.L.); 4Department of Nephrology, University of Helsinki and Helsinki University Hospital, 00290 Helsinki, Finland; 5Clinical and Molecular Metabolism, Faculty of Medicine Research Programs, University of Helsinki, 00014 Helsinki, Finland; 6Medical Microbiology and Immunology, Research Unit of Biomedicine, University of Oulu, 90014 Oulu, Finland; sohvi.horkko@oulu.fi; 7Medical Research Center Oulu, Oulu University Hospital and University of Oulu, 90014 Oulu, Finland; 8Department of Diabetes, Central Clinical School, Monash University, Melbourne, VIC 3004, Australia; 9Department of Clinical Medicine, University of Copenhagen, 2200 Copenhagen, Denmark

**Keywords:** type 1 diabetes, intestinal inflammation, albuminuria, butyrate, intestinal alkaline phosphatase

## Abstract

Type 1 diabetes is associated with increased intestinal inflammation and decreased abundance of butyrate-producing bacteria. We investigated the effect of butyrate on inflammation, kidney parameters, HbA1c, serum metabolites and gastrointestinal symptoms in persons with type 1 diabetes, albuminuria and intestinal inflammation. We conducted a randomized placebo-controlled, double-blind, parallel clinical study involving 53 participants randomized to 3.6 g sodium butyrate daily or placebo for 12 weeks. The primary endpoint was the change in fecal calprotectin. Additional endpoints were the change in fecal short chain fatty acids, intestinal alkaline phosphatase activity and immunoglobulins, serum lipopolysaccharide, CRP, albuminuria, kidney function, HbA1c, metabolites and gastrointestinal symptoms. The mean age was 54 ± 13 years, and the median [Q1:Q3] urinary albumin excretion was 46 [14:121] mg/g. The median fecal calprotectin in the butyrate group was 48 [26:100] μg/g at baseline, and the change was −1.0 [−20:10] μg/g; the median in the placebo group was 61 [25:139] μg/g at baseline, and the change was −12 [−95:1] μg/g. The difference between the groups was not significant (*p* = 0.24); neither did we find an effect of butyrate compared to placebo on the other inflammatory markers, kidney parameters, HbA1c, metabolites nor gastrointestinal symptoms. Twelve weeks of butyrate supplementation did not reduce intestinal inflammation in persons with type 1 diabetes, albuminuria and intestinal inflammation.

## 1. Introduction

An interesting interplay between gut microbes and host has been discovered in general as well as in diabetes, and the search for the central link between the gastrointestinal milieu and the development of diabetic complications has begun [1]. Inflammation is a key player in the development of diabetic complications, and inflammation in the gut has probable consequences beyond the gastrointestinal tract. Interestingly, fecal calprotectin, a marker of neutrophil activity, is increased in persons with type 1 diabetes, indicating a possible subclinical inflammatory state [2]. This might be related to disturbance of the bacterial composition, with a shift to lower bacterial diversity and less butyrate-producing bacteria, in persons with type 1 diabetes [3]. The gut microbiota has been suggested as a possible modulator of type 1 diabetes onset, and increased intestinal permeability in combination with intestinal dysbiosis is evident around diabetes onset in children [4,5]. The cause of this bacterial imbalance in diabetes is unknown, and the systemic effects of subclinical gut inflammation in persons with type 1 diabetes are still to be investigated.

It has been demonstrated that the short chain fatty acid (SCFA) butyrate produced by colonic bacteria—mainly *Firmicutes*—by fermentation of fibers, exerts beneficial metabolic, anti-inflammatory and anti-carcinogenic effects in epithelial cells, immune cells and adipocytes [6,7,8,9]. The anti-inflammatory effect is essential for maintaining the intestinal mucosal defense that segregates commensal and pathogenic bacteria, e.g., by upregulation of intestinal alkaline phosphatase (IAP) gene expression [10,11]. The brush-border enzyme IAP plays an important role by suppressing inflammatory mediators, including microbial compounds (e.g., endotoxins and polyphosphates) and luminal ATP by dephosphorylation [12,13]; IAP also increases immunoglobulin (Ig) A, which is crucial for the mucosal host–microbiota interplay, demonstrated in a mouse model [2,14]. Decreased luminal IAP activity, which has been reported in type 1 diabetes [2], possibly increases gut permeability and translocation of toxic proinflammatory compounds, e.g., lipopolysaccharides (LPS), into the intestinal vasculature, which can induce a systemic inflammatory response and thereby increase the risk of kidney or other organ injury [15,16]. An impaired intestinal barrier and an altered bacterial composition are also present in chronic kidney disease [17]; the causation is probably bidirectional, as accumulating uremic toxins are responsible for integrity loss in the intestinal barrier [18]. In a rat study, induction of kidney disease was associated with loss of intestinal tight junctions and colonic mucin, simultaneous with an increase in circulating LPS; supplementation with butyrate improved the kidney function, strengthened the intestinal barrier and reduced the circulating LPS [19]. The kidney-protective effects of SCFA have also been demonstrated in animal models of diabetic kidney disease and acute kidney injury, via local and systemic anti-inflammatory actions [20,21]. In humans, butyrate supplement has shown beneficial effects in persons with various gastrointestinal conditions, e.g., irritable bowel syndrome and inflammatory bowel disease (IBD) [22,23]. These results indicate that early identification and proper management of gastrointestinal inflammation in persons with type 1 diabetes prevents the development of inflammation-mediated systemic disease. Butyrate’s effect on glucose metabolism has been tested in persons with type 2 diabetes, in whom it increased glucagon-like peptide 1 concentration, compared to a placebo [24]. Whether or not this effect is also present in type 1 diabetes is unknown. To our knowledge, butyrate’s effect on intestinal inflammation, IAP activity, kidney parameters, metabolites and HbA1c has never been tested in type 1 diabetes. We hypothesized that oral supplementation of butyrate for 12 weeks would have anti-inflammatory effects locally, resulting in decreased fecal calprotectin, but also systemically, thereby reducing albuminuria and HbA1c and improving kidney function in persons with type 1 diabetes, albuminuria and intestinal inflammation.

## 2. Materials and Methods

This dual-center, double-blinded, parallel randomized controlled trial included participants from the Steno Diabetes Center Copenhagen (SDCC), Denmark and the Folkhälsan Research Center Helsinki, Finland. The intervention period was from 23 August 2019 to 29 December 2020. The study was registered on 29 August 2019 at clinicaltrials.org (NCT04073927). All participants gave written informed consent. The study was approved by the Regional Ethics Committee in Denmark (26 March 2019, ID: H-18062027) and the Ethics Committee of the Helsinki and Uusimaa Hospital District in Finland and conducted in accordance with the Declaration of Helsinki. Adults with type 1 diabetes and a history of or present albuminuria (two out of three consecutive measurements of urinary albumin creatinine ratio (UACR) above 30 mg/g or 30 mg albumin in 24-h urine collections) were invited for screening. Inclusion required fecal calprotectin ≥ 50 µg/g, determined at home by PreventID CalDetect 50/200 (Preventis GmbH, Bensheim, Germany). Participants were given detailed oral and written test instructions. If the home test was without a visible control line, it was judged as invalid, and the participant was given a second test. If the second test also was without a visible control line, the participant was excluded (*n* = 1). The home test was only used for screening, and not for assessment of fecal calprotectin levels at baseline and end-of-study (EOS). Key exclusion criteria were the presence of IBD, symptoms of IBD (this was justified based on clinical assessment and, if the investigator was in doubt, after discussion with a gastroenterologist), celiac disease, estimated glomerular filtration rate (eGFR) < 15 mL/min/1.73 m^2^, dialysis or kidney transplantation, non-diabetic chronic kidney disease, systemic anti-inflammatory therapy, antibiotic therapy within 30 days, pregnancy or lactation. Participants were randomized in blocks of four by the Capital Region Pharmacy, Herlev, Denmark to receive capsules with granular sodium butyrate (1.8 g, corresponding to 6 capsules, twice daily) coated with sodium alginate to ensure delayed release, or matching placebo (microcrystalline cellulose) produced by Sensilab d.o.o., Ljubljana, Slovenia for 12 weeks. Participants, investigators and outcome assessors were all blinded to group assignment. Adherence was predefined as a returned capsule count by the investigator of maximum 20% at EOS. Antibiotic treatment initialized during the intervention period did not cause discontinuation but was registered for sensitivity analysis.

### 2.1. Study Procedures

The UACR was calculated as the geometric mean of three morning urine samples at baseline and EOS. Blood sampling and measurements of height, weight and blood pressure were obtained at baseline and EOS. Information on demographic data and medical history was collected by interview and from medical records. Fecal samples at baseline, week four, week eight and week twelve (EOS) were collected and immediately frozen at home to −18 degrees Celsius. For the Danish participants, samples were delivered to the site of investigation a maximum 48 h after collection, and transported in provided equipment to avoid thawing, before storing at −80 degrees Celsius. In Finland, participants submitted their fecal samples by mail, and samples were stored at −80 degrees Celsius after a median of 46 [25:64] hours after collection. Assisted questionnaires were completed at baseline and EOS, for evaluation of changes in gastrointestinal symptoms.

### 2.2. Endpoints

The primary endpoint was change in fecal calprotectin. Secondary endpoints included change in fecal IAP activity, SCFAs, UACR, kidney function and HbA1c. Tertiary endpoints were change in serum lipopolysaccharide (LPS), high sensitivity-C-reactive protein (hs-CRP), fecal immunoglobulins (Ig), 38 selected serum metabolites (Appendix A) and gastrointestinal symptoms.

### 2.3. Analyses

The eGFR was estimated using the Chronic Kidney Disease Epidemiology Collaboration (CKD-EPI) equation. Commercial Enzyme-linked Immunosorbent Assay (ELISA) (Bühlmann, Schönenbuch, Switzerland) was used for precise determination of fecal calprotectin concentrations. Fecal SCFA (acetate, propionate, butyrate and valerate) concentrations were quantitated by gas chromatography-mass spectrometry. For the analyses of IAP and immunoglobulins, fecal samples (50 mg) were homogenized in 500 µL extraction buffer (0.1 mM ZnCl2, 1 mM MgCl2, 10 mM Tris–HCl, pH 8.0) using 0.1-mm glass beads (Precellys, Bertin Technologies, Montigny, France). After centrifugation (16,000× *g*, 10 min, +4 °C), supernatants were collected for the downstream analyses of fecal IAP activity levels, immunoglobulins (IgA, IgG and IgM) and protein concentrations as described earlier [2]. Serum LPS activities were determined with a Limulus Amebocyte Lysate (LAL) assay on 1:5 diluted samples in accordance with the manufacturer’s instructions (HyCult Biotech, Uden, The Netherlands). A panel of 38 metabolites, including bile acids, amino acids and other compounds, was quantified in serum using a targeted platform based on ultra-high-performance liquid chromatography coupled to mass spectrometry, as described previously [25]. The metabolite panel included compounds recognized as predictors of diabetes-related complications: among them, creatinine, gama-butyrobetaine, beta-hydroxybutyrate, N-methylnicotinamide, kynurenine, leucine, isoleucine, phenylalanine, azelaic acid, tryptophan, asymmetric dimethyl arginine, amino adipic acid, taurine, trimethylamine-N-oxide, glycine, glutamine, glutamic acid, tyrosine and indoxyl sulfate. Metabolites were excluded if the concentration (ng/mL) was below the quantification limit in more than 70% of the samples.

### 2.4. Statistical Analysis

The sample size calculation was based on 80% power to detect a treatment effect of >10% reduction in fecal calprotectin at a significance level of 0.05, which required a sample size of a minimum 42 participants. To allow for dropouts, we included 53 participants. Categorical variables are reported as numbers (%). Continuous variables are presented as mean ± standard deviation (SD) if normal distributed, and the non-normal distributed variables are presented as median [lower quartile (Q1):upper quartile (Q3)] and transformed with a natural logarithm before analysis. The effect of butyrate compared to placebo on the EOS level with baseline value included as covariate was tested with the analysis of covariance (ANCOVA). According to the prespecified analysis plan, the main analyses evaluated the effect of butyrate compared to placebo after 12 weeks of treatment. Due to different fecal collection methods in Denmark and Finland, we tested for site-by-treatment interaction. The proportions of participants with a specific gastrointestinal symptom at baseline who experienced improvement were calculated, and proportions of all participants who experienced worsening or onset of a symptom were calculated for interpretation. In supplementary analysis we: (1) restricted the analyses of calprotectin and UACR to participants with elevated levels (>50 µg/g with ELISA test and >30 mg/g, respectively) at baseline (*n* = 29 and *n* = 30, respectively); (2) excluded participants who received antibiotics during the treatment period from analysis of changes in inflammatory parameters (*n* = 2); (3) excluded participants with dosage change in renin-angiotensin inhibitor (RASi) treatment during the study for the analysis of change in UACR (*n* = 3); (4) included calprotectin measurements from all four timepoints in a constrained linear mixed model with random intercept. Results are presented for the “intention-to-treat population”, that consisted of all randomized subjects, but an analysis restricted to adherent participants was also performed for the primary and secondary endpoints. All analyses were performed using SAS statistical software version 7.1 (SAS Institute Inc, Cary, NC, USA). A two-sided *p* value < 0.05 was considered significant.

## 3. Results

### 3.1. Baseline

A total of 131 persons consented to participate. Twelve dropped out before the calprotectin home test or were excluded after performing two invalid tests. Of the 119 persons who performed the calprotectin test correct at home, 56 (47%) had <50 μg/g, 48 (40%) had 50–200 μg/g and 15 (13%) had >200 μg/g calprotectin. The two latter groups were invited for randomization, and all 53 persons were randomized (28 butyrate, 25 placebo). One participant terminated the intervention after a serious adverse event (SAE) (ischemic stroke), and seven other participants were non-adherent. Two participants dropped out after week eight, and fecal samples from week eight were used as EOS samples for these two persons in the main analysis of the fecal endpoints, whereas they were not included in analyses of the other endpoints. A flow diagram is presented in Figure 1. Baseline characteristics are displayed in Table 1. Among the 53 individuals, 23 (43%) were women, mean ± SD age was 54 ± 13 years and diabetes duration was 30 ± 15 years. At baseline, median [Q1:Q3] UACR was 46 [14:21] mg/g with 21 (40%), 24 (45%) and 8 (15%) participants having normo-, micro- and macroalbuminuria, respectively. Clinical characteristics at baseline were balanced between the butyrate and placebo group, apart from current smokers, who were all in the placebo group, and treatment with metformin and statin was more frequent in the butyrate group. We found no cases of subclinical celiac disease or IBD at baseline. Only 29 (55%) of the participants had elevated fecal calprotectin (≥50 μg/g) at baseline, despite a positive calprotectin test at home. When all four fecal calprotectin measurements during the study were taken into consideration, 42 (79%) participants had fecal calprotectin value ≥ 50 μg/g at minimum one timepoint during the study period.

### 3.2. Changes in Fecal and Circulating Markers of Inflammation

None of the tested fecal biomarkers (calprotectin, IAP, immunoglobulins, SCFAs) changed significantly after butyrate supplementation, compared to placebo (Table 2). From a baseline median [Q1:Q3] calprotectin level of 46 [26:100] μg/g, the median change in the butyrate group was −1.0 [−20:10] μg/g, and from a baseline median level of 61 [25:139] μg/g, the median change was −12 [−95:1] μg/g in the placebo group, with no difference between treatment groups (*p* = 0.24). Fecal LPS and serum hs-CRP did not change with butyrate supplementation compared to placebo (*p* = 0.13 and 0.093, respectively) (Table 2). Analyses restricted to participants with calprotectin measured by ELISA of minimum 50 µg/g at baseline (*n* = 29) showed a significant difference in the effect on calprotectin (*p* = 0.044) between the treatment groups, driven by a substantial decrease in the placebo group of median −44 [−151:−3] μg/g from a baseline level of 133 [62:304] μg/g, compared to a decrease in the butyrate group of −13 [−50:73] μg/g from a baseline level of 100 [72:229] μg/g. The other results were confirmatory. We found no site-by-treatment interaction (*p* ≥ 0.20) for any of the fecal markers. We checked that the protein normalized results (IAP and immunoglobulins) were not biased by a change in fecal protein concentrations. No change in fecal protein concentration was demonstrated (*p* for paired *t*-test = 0.59 and 0.43 in the butyrate and placebo group, respectively). No indication of a temporary effect on fecal parameters was demonstrated when we visualized the results from all four timepoints of the study (Figure 2).

### 3.3. Changes in Kidney Parameters, HbA1c and Metabolites

The UACR did not change with butyrate supplementation compared to placebo (Table 2). From a median baseline UACR level of 39 [18:121] mg/g, the median change was 1.5 [−9.5:21] mg/g after butyrate and −1.0 [−19:6.0] mg/g after placebo from a baseline of 49 [14:121] mg/g (*p* = 0.69 for difference between groups). For eGFR, the mean (SD) change was −2.0 (8.3) mL/min/1.73 m^2^ from a baseline of 86 (26) mL/min/1.73 m^2^ for butyrate and 0.39 (7.1) mL/min/1.73 m^2^ from a baseline of 82 (22) mL/min/1.73 m^2^ for placebo (*p* = 0.30, for difference between groups). Analyses restricted to the 30 participants with albuminuria at the baseline visit were consistent with the main analysis and did not show any significant effect on the UACR (*p* = 0.96, for difference between groups). There was no significant change in HbA1c, which was mean 0.1 (0.4)% (mmol/mol) in the butyrate group and 0.2 (0.6)% (mmol/mol) in the placebo group (*p* = 0.32 for difference between groups). Out of 38 metabolites, 8 were excluded due to non-detectable values for more than 70% of the samples. None of the 30 metabolites were changed after butyrate supplementation compared to placebo (Appendix A).

### 3.4. Changes in Gastrointestinal Symptoms

The proportions of participants who experienced changes in gastrointestinal symptoms in the two treatment groups are presented in Appendix A. No convincing effect of butyrate supplementation in any direction could be demonstrated: improvement, as well as worsening and onset of gastrointestinal symptoms during the study period, was reported in both the butyrate and in the placebo group. The numbers of participants who experienced changes in gastrointestinal symptoms were too small for statistical testing.

### 3.5. Supplementary Analysis

The results were consistent after the two participants that received antibiotics during the treatment period were excluded from the analyses of the inflammatory parameters; and after the three participants with change in RASi dosage during the study were excluded from the analyses of the UACR. The results of the analysis only including the 43 adherent participants (excluding 8 non-adherent participants and two dropouts) were confirmatory. We did not find a difference between the effect of butyrate and placebo on fecal calprotectin when including all four measurements in a mixed model (*p* = 0.31).

### 3.6. Safety

Eighteen adverse events were registered in the butyrate group, and 16 in the placebo group. Eight SAEs were reported. including: one death before randomization; one debut of multiple sclerosis; one ischemic stroke; one worsening of angina leading to elective percutaneous transluminal coronary angioplasty; one lower limb arterial thrombosis in the group receiving butyrate; one thrombosis and non-ST segment elevation myocardial infarction following a planned femoral bypass; and one hospitalization due to an infected diabetic leg ulcer in the placebo group.

## 4. Conclusions

In this study including persons with type 1 diabetes, intestinal inflammation and albuminuria, we could not demonstrate any effect on inflammatory markers, SCFAs, kidney parameters, HbA1c or selected metabolites after 12 weeks of butyrate supplementation. Furthermore, no convincing effect on gastrointestinal symptoms was found. Adherence was assessed by returned capsule count, and 43 of the 50 persons (86%) who completed the study met our pre-specified adherence criterion of a maximum 20% returned capsules, even though the daily dose counted as many as 12 capsules split in two doses. To our knowledge, this is the first study to evaluate the effect of butyrate supplementation on gut inflammation and diabetic nephropathy in type 1 diabetes.

Butyrate supplementation has beneficial effects on IBD, which shares genetic and immunological aspects with type 1 diabetes [26]. Both multifactorial autoimmune diseases are associated with increased risk of cardiovascular disease and premature mortality [27,28], and the prevalence of IBD is approximately 6-fold higher in adults with type 1 diabetes than in non-diabetic adults [29]. Interestingly, our finding of increased calprotectin level in 53% of the screened persons indicates a subclinical inflammatory state in this type 1 diabetes population, even when the possible overestimation by the calprotectin home test and the risk of selection bias is considered. Butyrate supplementation (of 4 g) as an add-on therapy showed significant reduction in fecal calprotectin after six months, compared to standard treatment in a population with ulcerative colitis and a median calprotectin level of 226 [167:309] μg/g [23]. Compared to our findings, one might reflect that butyrate is only beneficial in severe intestinal inflammation.

As butyrate is an energy source for colonocytes, the excretion rate is only 5–10% and is furthermore affected by transit time and diet, which makes the inter- and intraindividual variability substantial [30,31]. The implication of fecal SCFA measurements is disputed as the correlation with colonic availability is unknown. We included SCFAs as endpoints to examine whether changes in SCFA were correlated to a potential change in calprotectin level. We did not find a change in fecal butyrate in the group treated with butyrate. In a recent clinical trial by de Groot et al., butyrate supplementation reduced fecal butyrate levels in subjects with type 1 diabetes. The authors speculate that butyrate supplementation may have a counterregulatory effect on the microbial butyrate production [32]. Several drugs affect the gut microbiota and inflammatory markers and are therefore possible confounders. Non-steroidal inflammatory drugs have been reported to increase fecal calprotectin, but also proton pump inhibitors are associated with increased fecal calprotectin [33]. These over-the-counter-drugs are often used intermittently, and treatment onset or termination may bias the results in both directions. Additionally, metformin and statins might increase the number of butyrate-producing bacteria [34,35], and there was a higher frequency of treatment with these drugs in the group treated with butyrate. This imbalance would potentially bias the results towards an overestimation of the true effect of butyrate.

We were not able to demonstrate that butyrate supplementation changed the IAP activity (which preserve the gut blood-barrier) nor the concentration of circulating LPS. Neither did butyrate supplementation promote a beneficial shift in gut-related metabolites. Besides the proinflammatory effect of endotoxins passing a defect gut barrier, the inflammatory cascade can also be triggered by microbiota-derived metabolites [36]. An individual’s metabolic profile is associated with development of diabetic complications including chronic kidney disease [37], but metabolomics has not previously been evaluated in relation to butyrate supplementation in humans.

This study was not primarily designed to evaluate changes in gastrointestinal symptoms, as the presence of symptoms was not an inclusion criterion. However, one explanation for our findings could be that the gastrointestinal symptoms in our population were due to neuropathy. We are not aware of previous investigations of the effect of butyrate on gastrointestinal symptoms in diabetes. Neither has the effect of butyrate on kidney parameters been tested, though an association between intestinal dysbiosis and diabetic kidney disease has been described in humans [38].

This randomized placebo-controlled trial could not demonstrate that butyrate supplementation is the loophole to intestinal health in persons with intestinal inflammation with the applied dose. The link between dysbiosis and intestinal inflammation is evident [39], but the causal relationships and methods to interfere with the human intestinal environment are not understood. A fiber rich diet, shifting the microbiome towards butyrate-producing bacteria may be ideal, but profound lifestyle changes are hard to realize [40]. Facilitating butyrate production by increasing availability of fibers and butyrate-producing bacteria could potentially mimic a healthier diet. Supplementation with combinations of pre- and probiotics, synbiotics, has already shown beneficial effects in different populations [41,42]. The next step in the anti-inflammatory pathway, IAP, has also been investigated as a treatment target. Exogenous IAP administration has shown beneficial effects on calprotectin level in ulcerative colitis, kidney function in acute kidney injury and inflammatory markers in both populations, but the enzyme is challenging to administer [43,44,45].The question is if a shortcut to improve intestinal environment in susceptible persons is within reach or if the only way forward is to feed the beneficial strains.

The design of this dual-center, double-blinded randomized placebo-controlled trial is a major strength. Intervention targeting the colon is pharmacologically challenging, as protection against gastric pH and enzymic degradation is crucial for orally administered products. In this study, capsules with delayed release were used to ensure the delivery of butyrate to the terminal ileum and colon. The compliance of the study participants may have been an issue because of the capsule quantity, counting 12 capsules daily. The daily dose was split in two, to ease ingestion. This regimen required compliant participants, and the mean baseline HbA1c level of 8.0% (mmol/mol) suggests that the participants did not follow a strict medical regimen in general. On the other hand, the information about an inflammatory condition that was given to the participants during inclusion might explain why 86% of the participants who completed the study were adherent based on returned capsule count with a prespecified threshold of 80%. We aimed to study the effect of butyrate in a population with intestinal inflammation. To facilitate the screening proces, we used a calprotectin home test for screening. Of the screened persons, 53% had increased fecal calprotectin measured by this test, and were randomized, but only 55% had increased fecal calprotectin measured with ELISA test at randomization. This discrepancy can be explained by the time lag (1−3 weeks) between screening and randomization, intra-individual variability and/or low specificity of the home test. We accounted for the unintended inclusion of persons with normal calprotectin level by performing supplementary analyses restricted to participants with elevated fecal calprotectin level at randomization, though this sample size reduction may have resulted in insufficient power. Only one person was excluded, due to two invalid home calprotectin tests, limiting the influence of pre-analytical errors. Guided by previous human studies, we used a dose of butyrate of 3.6 g/day, but this might be an ineffective dose, as beneficial effects have been demonstrated with 10-times higher butyrate doses in studies with domestic animals [46].

In conclusion, in this randomized placebo-controlled trial of persons with type 1 diabetes, a butyrate supplement of 3.6 g/day was without adverse events, but also unable to change fecal and circulating inflammatory markers, SCFAs, kidney parameters, Hba1c, selected metabolites and gastrointestinal symptoms within 12 weeks. Even though butyrate plays a central role in the intestinal dysbiosis that is evident in both type 1 diabetes and diabetic kidney disease, a supplement of this bacterial product is not a shortcut to rebalance the gut microbiome in type 1 diabetes.

## Figures and Tables

**Figure 1 jcm-11-03573-f001:**
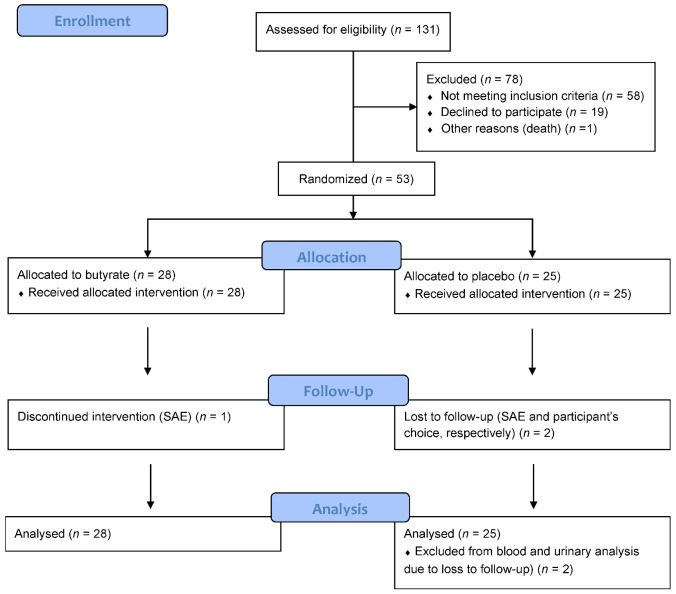
Flow diagram.

**Figure 2 jcm-11-03573-f002:**
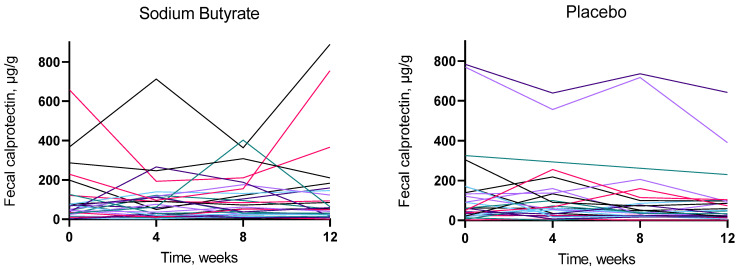
Effects of 12 weeks of treatment with sodium butyrate or placebo on fecal calprotectin. Each colored line represents one participant.

**Table 1 jcm-11-03573-t001:** Baseline characteristics.

	Total	Sodium Butyrate	Placebo
Number	53 (100)	28 (53)	25 (47)
Site (Denmark)	41 (77)	22 (79)	19 (76)
Sex (Female)	23 (43)	11 (39)	12 (48)
Age, years	54 ± 13	56 ± 11	52 ± 15
Diabetes duration, years	30 ± 15	29 ± 17	32 ± 14
Current smoking	7 (13)	0	7 (28)
Body Mass Index, kg/m^2^	29 ± 5.8	30 ± 6.1	29 ± 5.7
Systolic blood pressure, mm Hg	135 ± 18	136 ± 15	134 ± 21
Diastolic blood pressure, mm Hg	78 ± 10	80 ± 10	77 ± 10
Urinary albumin creatinine ratio, mg/g	46 [14:121]	39 [18:121]	49 [14:121]
Normoalbuminuria	21 (40)	11 (39)	10 (40)
Microalbuminuria	24 (45)	13 (46)	12 (44)
Macroalbuminuria	8 (15)	4 (14)	3 (12)
eGFR, mL/min/1.73 m^2^	84 ± 24	86 ± 26	8.1 ± 1.1
HbA_1c_, % (mmol/mol)	8.0 ± 1.0	8.0 ± 0.8	8.1 ± 1.1
Fecal calprotectin ≥ 50 µg/g	29 (55)	14 (48)	15 (52)
RASi treatment	44 (83)	23 (82)	21 (84)
Statin treatment	37 (70)	21 (75)	16 (64)
Metformin treatment	10 (19)	7 (25)	3 (12)

Data are shown as number (%), mean ± standard deviation, or median [Q1:Q3]. eGFR indicates estimated glomerular filtration rate; RASi, renin-angiotensin system inhibitor.

**Table 2 jcm-11-03573-t002:** Change in fecal, blood, urinary markers and HbA1c after 12 weeks intervention, compared by ANCOVA.

	Baseline	After Intervention	Change	ANCOVA *p* Value
Fecal calprotectin (µg/g)				
Sodium butyrate	48 [26:100]	50 [19:135]	−1.0 [−20:10]	0.24
Placebo	61 [25:139]	47 [19:95]	−12 [−95:1]	
Fecal IAP activity/Protein (U/g)				
Sodium butyrate	92 [49:609]	83 [39:605]	−3.9 [−25:51]	0.98
Placebo	55 [38:133]	53 [22:106]	−0.24 [−22:38]	
Fecal butyrate (µg/mg)				
Sodium butyrate	6.4 [2.8:12]	6.4 [3.7:12]	0.7 [−1.5:4.2]	0.34
Placebo	4.3 [2.1:7.6]	4.4 [2.5:8.8]	0.052 [−3.5:2.0]	
Fecal acetate (µg/mg)				
Sodium butyrate	21 ± 8.8	21 ± 12	0.080 ± 12	0.33
Placebo	18 ± 9.6	17 ± 8.7	−1.3 ± 10	
Fecal propionate (µg/mg)				
Sodium butyrate	8.0 [4.2:12]	6.8 [3.5:11]	−0.15 [−3.3:1.1]	0.93
Placebo	5.4 [3.6:7.9]	5.5 [3.7:8.7]	−0.94 [−2.3:1.8]	
Fecal valerate (µg/mg)				
Sodium butyrate	2.0 ± 1.0	2.1 ± 1.6	0.13 ± 1.8	0.67
Placebo	1.7 ± 1.0	1.8 ± 1.2	0.15 ± 1.3	
Fecal immunoglobulin G/Protein (ng/g)				
Sodium butyrate	28 [16:110]	40 [16:116]	−1.8 [−0.067:0.034]	0.10
Placebo	27 [14:67]	17 [9.9:36]	−1.9 [−15:18]	
Fecal immunoglobulin A/Protein (µg/mg)				
Sodium butyrate	1.6 [0.50:7.14]	2.3 [0.72:4.9]	0.30 [−1.2:1.31]	0.69
Placebo	2.4 [1.1:7.8]	1.7 [0.73:9.9]	0.077 [−2.0:3.7]	
Fecal immunoglobulin M/Protein (ng/mg)				
Sodium butyrate	0.61 [0.32:2.0]	1.1 [0.36:2.7]	0.38 [−0.0011:1.5]	0.20
Placebo	1.0 [0.36:1.9]	0.72 [0.46:1.8]	0.18 [−1.4:0.70]	
Urinary albumin creatinine ratio (mg/g)				
Sodium butyrate	39 [18:121]	37 [27:82]	1.5 [−9.5:21]	0.69
Placebo	49 [14:121]	37 [18:208]	−1.0 [−19:6.0]	
Estimated glomerular filtration rate (mL/min/1.73 m^2^)				
Sodium butyrate	86 ± 26	84 ± 25	−2.0 ± 8.3	0.30
Placebo	82 ± 22	83 ± 23	0.39 ± 7.1	
Serum high sensitivity CRP (mg/L)				
Sodium butyrate	1.7 [0.91:3.1]	1.7 [0.95:3.0]	−0.06 [−0.65:0.55]	0.13
Placebo	1.9 [1.0:4.0]	2.0 [1.5:6.0]	−0.18 [−0.84:0.38]	
Serum lipopolysaccharide (EU/mL)				
Sodium butyrate	0.66 ± 0.18	0.66 ± 0.20	−0.0025 ± 0.20	0.093
Placebo	0.73 ± 0.11	0.80 ± 0.26	0.076 ± 0.26	
HbA1c (% (mmol/mol))				
Sodium butyrate	8.0 ± 0.8	8.0 ± 0.9	0.1 ± 0.4	0.32
Placebo	8.1 ± 1.1	8.3 ± 1.3	0.2 ± 0.6	

Values are mean ± standard deviation, median [Q1:Q3]. Parameters with a non-normal distribution were log transformed before analysis. *p* value for the group-wise comparison of participants treated with sodium butyrate or placebo was calculated using baseline-corrected linear regression.

## Data Availability

Data is available upon reasonable request to the corresponding author.

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
