# Peer review of "Effects of Butyrate Supplementation on Inflammation and Kidney Parameters in Type 1 Diabetes: A Randomized, Double-Blind, Placebo-Controlled Trial"

_jcm, 2022, doi:10.3390/jcm11133573_

Round 1

Reviewer 1 Report

The authors present a comprehensive study on the effects of butyrate supplementation in patients with Type I diabetes. Primary endpoint is changes in fecal calprotectin, additional endpoints are fecal SCFA, inflammation maker, intestinal symptoms and kidney function markers. No significant differences were identified in parameters.

Microbiome derived butyrate and other SCFAs are often associated with improved gut health in a number of GI disorders (IBS, IBD) and reduction in inflammation marker, calprotectin. This study demonstrated that GI inflammation and symptoms did not respond to butyrate. The outcomes of this study is important in improving our understanding of gut-butyrate interactions and its effect in diabetic patients. 

No additional changes required.    

Author Response

Thank you for reviewing this manuscript and for your kind comments.

Reviewer 2 Report

The authors provide an interesting and sound article and although the clinical intervention resulted in no obvious effect, it is valuablke for a specific readership.

The introduction has a very focused view on the molecular mechanisms of IAP, but in contrast, the hypothesis of this study is then leading to a rather systemically alteration of glucose metabolism (e.g. change in HbA1c). How do the authors justify regarding their hypothesis, the possible effect of butyrate on energy metabolism and/or diabetic nephropathy?

Please discuss the preanalytical failures which can happen when calprotectin is determined by study participants at home. Within this context, please specify what was judged as two invalid calprotectin tests.

Please specify “exclusion criteria were the presence of IBD, symptoms of IBD (justified by the investigators)” (line 91-92)

Did the treatment/ingestion of butyrate lead to any bowel problems such as diarrhea or comparable?

Given the background that the T1D patients showed rather high baseline levels of HbA1c, please discuss the problem of therapeutic compliance when trying to initiate intervention studies with participants not following the strict medical regimen in general.

Please discuss, why there are no changes in hydroxybutyric acid despite the intervention of butyrate supplementation over 12 weeks. Is it possible the butyrate was not resorbed in the gut? Or used as an energy source for microbiota?

Line 293: “Adherence was assessed by capsule count and 43 of the 50 persons (86%) who completed….”

What does that mean? Please specify and discuss that in more depth. It could be also possible that the adherence/compliance was biased by multiple effects (e.g. 15 capsules at once etc.). This is a major
